# Synergistic Roles of Non-Homologous End Joining and Homologous Recombination in Repair of Ionizing Radiation-Induced DNA Double Strand Breaks in Mouse Embryonic Stem Cells

**DOI:** 10.3390/cells13171462

**Published:** 2024-08-30

**Authors:** Gerarda van de Kamp, Tim Heemskerk, Roland Kanaar, Jeroen Essers

**Affiliations:** 1Department of Molecular Genetics, Erasmus MC Cancer Institute, Erasmus University Medical Center, 3015 GD Rotterdam, The Netherlands; 2Oncode Institute, Erasmus University Medical Center, 3015 GD Rotterdam, The Netherlands; 3Department of Vascular Surgery, Erasmus University Medical Center, 3015 GD Rotterdam, The Netherlands; 4Department of Radiotherapy, Erasmus MC Cancer Institute, Erasmus University Medical Center, 3015 GD Rotterdam, The Netherlands

**Keywords:** DNA double strand break, DNA-PK_cs_, DNA repair, homologous recombination, ionizing radiation, mouse embryonic stem cells, non-homologous end joining, Rad54

## Abstract

DNA double strand breaks (DSBs) are critical for the efficacy of radiotherapy as they lead to cell death if not repaired. DSBs caused by ionizing radiation (IR) initiate histone modifications and accumulate DNA repair proteins, including 53BP1, which forms distinct foci at damage sites and serves as a marker for DSBs. DSB repair primarily occurs through Non-Homologous End Joining (NHEJ) and Homologous Recombination (HR). NHEJ directly ligates DNA ends, employing proteins such as DNA-PK_cs_, while HR, involving proteins such as Rad54, uses a sister chromatid template for accurate repair and functions in the S and G2 phases of the cell cycle. Both pathways are crucial, as illustrated by the IR sensitivity in cells lacking DNA-PK_cs_ or Rad54. We generated mouse embryonic stem (mES) cells which are knockout (KO) for DNA-PK_cs_ and Rad54 to explore the combined role of HR and NHEJ in DSB repair. We found that cells lacking both DNA-PK_cs_ and Rad54 are hypersensitive to X-ray radiation, coinciding with impaired 53BP1 focus resolution and a more persistent G2 phase cell cycle block. Additionally, mES cells deficient in DNA-PK_cs_ or both DNA-PK_cs_ and Rad54 exhibit an increased nuclear size approximately 18–24 h post-irradiation. To further explore the role of Rad54 in the absence of DNA-PK_cs_, we generated DNA-PK_cs_ KO mES cells expressing GFP-tagged wild-type (WT) or ATPase-defective Rad54 to track the Rad54 foci over time post-irradiation. Cells lacking DNA-PK_cs_ and expressing ATPase-defective Rad54 exhibited a similar phenotypic response to IR as those lacking both DNA-PK_cs_ and Rad54. Despite a strong G2 phase arrest, live-cell imaging showed these cells eventually progress through mitosis, forming micronuclei. Additionally, mES cells lacking DNA-PK_cs_ showed increased Rad54 foci over time post-irradiation, indicating an enhanced reliance on HR for DSB repair without DNA-PK_cs_. Our findings underscore the essential roles of HR and NHEJ in maintaining genomic stability post-IR in mES cells. The interplay between these pathways is crucial for effective DSB repair and cell cycle progression, highlighting potential targets for enhancing radiotherapy outcomes.

## 1. Introduction

DSBs are an important determinant of the effectiveness of radiotherapy since they lead to cell death when left unrepaired. The induction of DSBs by IR results in the modification of histones, which ultimately leads to the accumulation of DNA repair proteins such as 53BP1 [1,2,3]. The assembly of 53BP1 into distinct foci at the damage site allows its utilization as a surrogate marker to visualize DSBs [3]. In addition, together with RIF1, PTIP, and the Shieldin complex, 53BP1 plays a critical role in guiding DSB repair by protecting DNA ends from end resection by proteins such as the MRN complex and CtIP [4]. Protected, blunt ends can be repaired by NHEJ, while resected ends can be repaired by HR.

Both repair pathways, NHEJ and HR, play a major role in the repair of DSBs. The NHEJ repair pathway commences with the accumulation of the Ku70-Ku80 heterodimer at the double-stranded DNA ends [5]. Subsequently, DNA-PK_cs_ binds to Ku70/80 and becomes active through autophosphorylation [6]. At the final stage of NHEJ, the XRCC4–ligase 4 complex directly ligates the DNA ends. NHEJ restores the structural integrity of the DNA but typically leads to short deletions and insertions. In contrast to NHEJ, HR is an accurate, high-fidelity repair pathway that employs the sister chromatid as a template to repair DSBs. Consequently, it operates exclusively in the S and G2 phases of the cell cycle. HR acts upon resected 3’ DNA overhangs, which are initially coated by the single-strand DNA binding protein RPA. Rad51, a central HR protein, displaces RPA and facilitates homologous DNA pairing and strand invasion [7,8]. The displacement of RPA by Rad51 is mediated by BRCA2. Rad54, a member of the Swi2/Snf2 family of DNA-dependent ATPases, binds to Rad51 and mediates the homology recognition and strand invasion [4,9,10]. The ATPase function of Rad54 is crucial for the disassembly of Rad54 from DNA, facilitating Rad54 turnover at DSBs and relocating the DSB repair sites within the nucleus [11]. However, the accumulation of Rad54 at the damage site is independent of its ATPase activity [11]. Rad54 also plays a role in catalyzing nucleosome remodeling and stabilizing the Rad51 filament at DSBs [9,10,12,13].

The importance of the NHEJ protein, DNA-PK_cs_, in the repair of DSBs in eukaryotic cells is illustrated by the fact that a loss of DNA-PK_cs_ in both mES cells and human tumor cells renders them highly sensitivity to IR [14,15]. Likewise, mES Rad54^−/−^ and chicken DT40 RAD54^−/−^ cells demonstrate sensitivity to IR, underscoring the significant role of HR in the repair of IR-induced DNA DSBs [16,17]. In animal models, DNA-PK_cs_ null mutations lead to the radiosensitive severe combined immunodeficiency (SCID) phenotype [18,19] Interestingly, while Rad54 KO adult mice do not display IR sensitivity, SCID Rad54^−/−^ mice exhibit increased IR sensitivity compared to SCID mice [20]. This phenomenon was also observed when deletions of Ku80 or Lig4 were combined with the deletion of Rad54 [21,22]. Additionally, fibroblasts derived from SCID mice expressing an ATPase-defective human RAD54 protein are hypersensitive to IR compared to single mutants [23]. Collectively, these findings have led to the conclusion that HR plays a primary role in non-differentiated cell types and acts as back-up for NHEJ in differentiated cell types. Meanwhile, NHEJ acts as a major pathway for the repair of DSBs induced by IR in all developmental stages.

To further understand the cooperating role of HR and NHEJ in the repair of IR-induced DSBs in eukaryotic cells, we generated mES cells which are KO for both DNA-PK_cs_ and Rad54. To study Rad54 and the role of its ATPase when DNA-PK_cs_ is absent, we generated mES DNA-PK_cs_ KO cells expressing either WT Rad54-GFP or ATPase-defective Rad54-GFP. We show that mES cells lacking both DNA-PK_cs_ and Rad54 or DNA-PK_cs_^−/−^ mES cells expressing ATPase-defective Rad54 are hypersensitive to X-ray radiation. Furthermore, these cells exhibit delayed resolution of 53BP1 foci, alongside an enlargement in nuclear size and extended G2 cell cycle arrest following irradiation. Moreover, there is increased micronuclei formation in cells deficient in HR and NHEJ after X-ray irradiation. These findings indicate that HR and NHEJ together play a major role in maintaining genomic stability in mES cells following IR treatment.

## 2. Materials and Methods

### 2.1. mES Cell Culture

Mouse embryonic stem (mES) (IB10, subclone of E14 129/Ola [24]) cell lines were maintained on 0.1% gelatin-coated dishes in DMEM (Gibco, Waltham, MA, USA) in a 1:1 mixture with Buffalo rat liver (BRL)-conditioned DMEM supplemented with 10% FCS, 1% non-essential amino acids (Lonza, Basel, Switzerland), 200 U/mL penicillin, 200 µg/mL streptomycin, 89 µM β-mercaptoethonal, and 1000 U/mL leukemia inhibitory factor. The Rad54^−/−^ [17], DNA-PK_cs_^−/−^ [15], IB10 Rad54^WT-GFP/−^, and IB10 Rad54^KR-GFP/−^ [11] mES cells were described before.

### 2.2. Generation of mES DNA-PK_cs_^−/−^ Rad54^−/−^ and DNA-PK_cs_^−/−^ Rad54^GFP^-Knockin Cell Lines

To generate the DNA-PK_cs_^−/−^ Rad54^−/−^ mES cells, DNA-PK_cs_^−/−^ cells were electroporated with an mRad54^307hyg^-targeting construct (described in [17]). Hygromycin (200 µg/mL) was added one day after electroporation, and the cells were maintained under selection for 8 days. DNA from hygromycin-resistant clones was isolated, and the clones were genotyped by PCR to select clones targeted at the Rad54 locus. To obtain DNA-PK_cs_^−/−^ mES cell lines that were homozygously knocked out for Rad54, an mRad54^307hyg-^targeted DNA-PK_cs_^−/−^ cell line was targeted with the mRad54^307pur^ construct (described in [17]). Puromycin (1.6 µg/mL) was added one day after electroporation, and the cells were maintained under selection for 8 days. DNA from puromycin-resistant clones was isolated, and clones were genotyped by PCR to select clones targeted at both alleles of Rad54.

To generate DNA-PK_cs_^−/−^ cells expressing GFP-tagged Rad54^WT^ or Rad54^K189R^, an mRad54^307hyg^-targeted DNA-PK_cs_^−/−^ cell line was electroporated with the Rad54^WT-GFP^ or the Rad54^K189R-GFP^ construct, which contains a puromycin resistance gene (described before in [11]). Replacement of the non-targeted Rad54 allele would generate cells with genotypes DNA-PK_cs_^−/−^ Rad54^WT-GFP/−^ and DNA-PK_cs_^−/−^ Rad54^KR-GFP/−^. One day after electroporation, puromycin (1.6 µg/mL) was added, and the cells were maintained under selection for 8 days. DNA was isolated from puromycin-resistant clones, and the clones were genotyped using PCR. Subsequently, the clones were sent for Sanger sequencing to confirm their genotype at amino acid 189 of the Rad54 sequence.

### 2.3. Western Blotting

The day before making the cell lysate, the cells were seeded and incubated overnight. To make the cell lysates, the cells were washed with cold PBS (Lonza), scraped from the dish, and incubated in Laëmmli buffer (4% SDS, 20% glycerol, 120 mM Tris) for 5 min at 95 °C to lyse the cells. The samples were sheared with a syringe to reduce their viscosity. To determine the protein concentration in the isolated protein samples, a Lowry assay was performed [25]. The samples were prepared by adding loading buffer (final concentration of 0.01% bromophenol blue and 0.5% β-mercaptoethanol). The samples were loaded onto SDS-page gel. The proteins were transferred onto an Odyssey Immobilin-P transfer membrane (Millipore, St. Louis, MI, USA). Blotting was performed for two hours at 4 °C at 300 mA in transfer buffer (0.4 M Glycine, 5 mM Tris, 20% methanol). After blotting, the membranes were incubated in blocking buffer (3% skim milk, 0.05% Tween-20 in PBS) for one hour at room temperature (RT). The membranes were incubated with the primary antibodies mouse anti-Rad54 (1:100, F-11 sc374598, Santa Cruz, San Francisco, CA, USA), mouse anti-Beta actin clone C4 (1:500,000, MAB1501R, Millipore), rabbit anti-vinculin (1:1000, NB129002, Abcam), and mouse anti-DNA-PK_cs_ (1:500, [18-2] ab1832-500, Abcam) in blocking buffer overnight at 4 °C. The membranes were washed with 0.05% Tween-20 in PBS. After washing, the membranes were incubated with secondary antibodies (1:2000, HRP anti-rabbit or HRP anti-mouse, Jackson ImmunoResearch Labs) for 1.5 h at RT. The membranes were washed, and after the addition of enhanced chemiluminescence substrate (homemade) to the blots, chemiluminescence was measured with an Amersham Imager 600 (GE Healthcare, Alpharetta, GA, USA).

### 2.4. Irradiation

All irradiations were performed using an Xstrahl RS320 X-ray generator with 195 kV X-rays (10 mA, 0.5 mm Cu filter).

### 2.5. Clonogenic Survival

The sensitivity of the mES cells to X-ray radiation was determined using a clonogenic survival assay. The mES cells were seeded at different cell densities onto 0.1% gelatin-coated 60 mm dishes. After 6–8 h, the cells were irradiated at the indicated dose or left untreated. After irradiation, the cells were grown for 7–8 days, fixed, and stained with Coomassie Blue staining solution (50% methanol, 7% acetic acid and 0.1% brilliant blue R). Colonies were counted with the GelCount colony counter (Oxford Optronix). To assess the relative sensitivity of the cell survival curves in comparison to each other, a linear-quadratic survival model was employed to fit the data points utilizing GraphPad Prism 9. Utilizing this curve fitting, the D37 values were derived and were used to calculate the fold change sensitivity.

### 2.6. Immunofluorescence Staining

Cells were seeded onto 18 mm coverslips in 6-well plates at a density of 300,000 cells/well. Before seeding, the coverslips were coated with 2.5 µg/mL Laminin (Roche) to ensure a monolayer of mES cells. The next day, the cells were irradiated, while the control samples were left untreated. At the indicated time points after irradiation, the cells were washed with PBS and fixed with 4% Paraformaldehyde (PFA). The cells were permeabilized with 0.1% Triton and subsequently blocked with PBS+ buffer (5 mg/mL Bovine Serum Albumin (BSA) and 1.5 mg glycine/mL PBS). The primary antibodies (53BP1; NB100-304; Novus Biologicals or phospho-H3; 06-570; Millipore, St. Louis, MI, USA) were diluted in PBS+ buffer (1:1000). The cells were incubated with the primary antibodies overnight at 4 °C. The cells were permeabilized with 0.1% Triton and washed with PBS+ buffer. Secondary antibodies (Anti-rabbit Alexa488 or Alexa594; Life Technologies, Carlsbad, CA, USA) were diluted in PBS+ buffer (1:1000). The cells were incubated in the dark for one hour at RT. Subsequently, the coverslips were mounted onto microscope slides using antifade mounting medium with DAPI (Vectashield). The coverslips were sealed with nail polish to prevent the samples from drying out.

### 2.7. Microscopy

To visualize the immunofluorescence in the cells, a Leica STELLARIS 5 or TCS SP5 confocal microscope was employed. The following laser lines were used: DAPI (405 nm laser), Alexa488 or GFP (488 nm laser), Alexa 594 (561 nm laser). For each sample, 5–8 z-stack images were captured using a 40× objective. Subsequently, z-projections were generated, and the nuclear area and mean and integrated density of the DAPI signal were measured for each nucleus. Additionally, the number of 53BP1 foci or Rad54 foci was analyzed for each nucleus. This was accomplished using homemade ImageJ scripts. In short, the cell nuclei were segmented based on DAPI signal, and the nuclear area and integrated density of the DAPI signal were quantified using the measurement function within ImageJ. For the identification of foci within the segmented nuclei, individual segmentation masks were created for each nucleus. Segmentation masks of the foci were generated using thresholds based on the mean + factor*standard deviation of the 53BP1 or Rad54 signal [26]. The number of segmented foci was then measured using the measurement function within ImageJ. The percentage of cells with micronuclei and the number of micronuclei per cell were quantified manually.

### 2.8. Live-Cell Imaging of Rad54-GFP

Time-lapse imaging was performed on a Leica SP8 confocal microscope. Fluorophores were excited with 488 nm (Rad54-GFP) and 638 nm lasers (Spy650-DNA), and the fluorescence was detected using a HyD detector. Using a 40× objective, z-stacks were acquired with a 1 μm slice thickness and a pixel size of 142 × 142 nm. The whole microscope was encased in an Okolab cage incubator to keep the samples at 37 °C and 5% CO_2_. One day prior to the experiment, 20,000 mES cells were seeded per well onto a μ-slide, 8-well chambered coverslip (Ibidi) that was coated with 2.5 µg/mL Laminin (Roche, Basel, Switzerland). The next day, the medium was refreshed with mES FluoroBrite medium (FluoroBrite DMEM (Gibco), 10% FCS, non-essential amino acids (Lonza), 200 U/mL penicillin, 200 μg/mL streptomycin, Ultraglutamine I (Lonza), 89 μM β-mercaptoethanol, and 1000 U/mL leukemia inhibitory factor with 1:10,000 Spy650-DNA dye (Spirochrome) one hour before irradiation. A z-stack of each position was made before irradiation. All the samples were irradiated with a 2 Gy X-ray and directly placed back into the microscope. Time-lapse imaging was started 10 min after irradiation with an interval of 30 min for 24 h. The images were analyzed using ImageJ image analysis software. A z-projection was made of all the z-stacks. The number of Rad54-GFP foci was analyzed using a homemade macro similar to that described for the immunofluorescent imaging. The percentage of cells undergoing division, cells with enlarged nuclei, and cells with micronuclei or chromatin bridges during division was quantified manually.

### 2.9. Cell Cycle Analysis

Cells were seeded onto 0.1% gelatin-coated 10 cm dishes and incubated overnight. Subsequently, the cells were irradiated with the indicated doses. The cells were incubated with 10 µM EdU (Invitrogen) for 30 min before fixation. The cells were trypsinized, pelleted, and fixed with 70% ethanol at the indicated time points after irradiation. For antibody and EdU staining, the cells were washed with 1% BSA in PBS after fixation. Next, the cells were permeabilized by incubating them with 0.1% Triton in PBS for 10 min on ice. Primary antibodies (phospho-H3, 06-570, Millipore) were diluted 1:500 in 0.1% Triton in PBS. The cells were incubated with the primary antibodies for two hours at RT. The cells were washed with 0.1% Triton in PBS and incubated for 30 min in the dark with the secondary antibody (anti-rabbit Alexa594, Life Technologies) diluted 1:1000 in 0.1% Triton in PBS. Subsequently, the cells were washed with 0.1% Triton in PBS, and EdU staining was performed using a Click-It EdU incorporation kit according to the manufacturer’s guidelines (Invitrogen, St. Bend, OR, USA). After EdU staining, the cells were washed with 1% BSA in PBS. Finally, the cells were incubated with PBS containing 1 µg/mL DAPI. If only DNA content staining was performed, the cells were washed with PBS after fixation and incubated with PBS containing 2 µg/mL Propidium Iodide (Invitrogen) and 0.25 µg/mL RNAse (Sigma-Aldrich) for 30 min at 37 °C. The fluorescence of the cells was measured using a BD Fortessa flow cytometer. The data were analyzed using the Flo-Jo software. Manual gating was performed on DNA content–EdU and DNA content–phospho-H3 graphs to determine the percentage of cells in the G1, S, G2, and mitotic phases. To calculate the percentage of cells in the G1, S, and G2 phases based on the DNA content histograms, the Dean–Jett–Fox model was employed [27].

### 2.10. Statistics

Statistical differences were calculated between the samples at each time point after irradiation to compare the number of 53BP1 or Rad54 foci, nuclear size, and the percentage of cells with micronuclei. Statistical differences were calculated using one-way ANOVA with the Šidák test to correct for multiple comparisons.

## 3. Results

### 3.1. mES Cells Lacking DNA-Pk_cs_ and Rad54 Are Hypersensitive to X-ray Radiation

We used gene targeting to inactivate Rad54 in the DNA-PK_cs_^−/−^ mES cell line. Two independent DNA-PK_cs_^−/−^ Rad54^−/−^ mES cell lines were generated (Figure 1a). We confirmed that the newly generated cell lines lacked protein expression of either DNA-PK_cs_ or Rad54 (Figure 1b). To investigate the sensitivity of these mES cells to IR, we subjected the WT, Rad54^−/−^, DNA-PK_cs_^−/−^, and Rad54^−/−^ DNA-PK_cs_^−/−^ mES cells to X-ray radiation and assessed their clonogenic survival. The Rad54 and DNA-PK_cs_ single KO mES cell lines showed, respectively, 2.0- and 2.6-fold increased sensitivity, based on the D37 values, compared to the WT mES cell line (Figure 1c, Appendix A). Interestingly, the DNA-PK_cs_^−/−^ Rad54^−/−^ mES cell lines were more sensitive than the single KO cell lines, showing a 5.2- and 6.3-fold increased sensitivity compared to the WT mES cell line (Figure 1c, Appendix A). These results show that cells that lack both DNA-PK_cs_ and Rad54 are hypersensitive to X-ray radiation.

### 3.2. mES Lacking DNA-PK_cs_ and Rad54 Cells Show Impaired 53BP1 Focus Resolution and an Increased Nuclear Size after 2 Gy of X-ray Radiation

To investigate whether the increased sensitivity of the DNA-PK_cs_^−/−^ Rad54^−/−^ mES cell lines is concurrent with the persistence of DSBs, we assessed the kinetics of 53BP1 focus formation and resolution after the induction of DSBs by X-ray radiation. We subjected the WT, Rad54^−/−^, DNA-PK_cs_^−/−^, and Rad54^−/−^ DNA-PK_cs_^−/−^ mES cells to 2 Gy of X-ray radiation and analyzed the number of 53BP1 foci in the untreated cells and at 2, 6, 8, 18, and 24 h after irradiation. We observed that the number of foci increased after X-ray irradiation and decreased over time in all genotypes of the mES cells. However, the decrease in 53BP1 foci over time was slower in both the Rad54^−/−^ and DNA-PK_cs_^−/−^ mES cell lines compared to the WT cells (Figure 2, Appendix A). Interestingly, the decrease in foci over time was even more impaired in the DNA-PK_cs_^−/−^ Rad54^−/−^ mES cells (Figure 2, Appendix A). At 2 h post-irradiation, the DNA-PK_cs_^−/−^ and DNA-PK_cs_^−/−^Rad54^−/−^ cells exhibited a higher number of 53BP1 foci compared to the WT and Rad54^−/−^ cells. For mES cells lacking DNA-PK_cs_ or Rad54, this difference was no longer apparent at 6 and 8 h after irradiation. These findings are consistent with the difference in the kinetics of repair by NHEJ and HR, with NHEJ typically completing within 30 min and HR taking 7 h or more [28,29]. The increased number of 53BP1 foci at 2 h after irradiation in mES cells lacking DNA-PK_cs_ likely results from impaired early DSB repair by NHEJ, whereas WT and Rad54^−/−^ cells are still capable of early repair through NHEJ. In the Rad54^−/−^ mES cells, which are deficient in HR, the delay in 53BP1 foci resolution becomes more pronounced at later time points. Consistent with these observations, mES cells deficient in both DNA-PK_cs_ and Rad54 exhibit a delay in 53BP1 foci resolution at both early and late time points after irradiation. Taken together, these results show that the hypersensitivity of DNA-PK_cs_^−/−^ Rad54^−/−^ mES cells to X-ray radiation coincides with the persistence of DSBs in these cells.

Furthermore, we observed that mES cells that lack both DNA-PK_cs_ and Rad54 show a substantial increase in nuclear area at 18 and 24 h post-irradiation (Figure 2, Appendix A). Compared to the WT mES cells, we observed an enlargement in the nuclear area in both the DNA-PK_cs_^−/−^ and DNA-PK_cs_^−/−^ Rad54^−/−^ mES cell lines. However, this increase in nuclear area was relatively modest in the DNA-PK_cs_^−/−^ mES cells, being only 1.2-fold larger than WT mES cells at 24 h after irradiation. In contrast, the DNA-PK_cs_^−/−^ Rad54^−/−^ mES cells exhibited a 2-fold increase in nuclear area compared to the WT mES cells. This observed enlargement in nuclear size suggests an elevated DNA content within these nuclei. Indeed, the DAPI integrated density per nucleus, which is a measure of the total DNA content and is defined as the nuclear area multiplied by the mean DAPI signal, is higher in the enlarged nuclei (> 300 µm^2^) (Appendix A). These results imply that cells with a large nuclear area might be arrested in their cell cycle progression after DNA replication, specifically in the G2 phase. This cell cycle arrest is probably caused by the presence of a high number of unrepaired DSBs since cells with a large nuclear area (>300 µm^2^) contain more 53BP1 foci compared to cells with a smaller nuclear area (Appendix A).

### 3.3. X-ray Irradiation Results in More Persistent G2 Phase Cell Cycle Block in mES Cells Lacking DNA-PK_cs_ and Rad54

To directly study the effect of X-ray radiation on cell cycle progression, we subjected WT, Rad54^−/−^, DNA-PK_cs_^−/−^, and Rad54^−/−^ DNA-PK_cs_^−/−^ mES cells to 1 and 2 Gy of X-ray radiation and analyzed cells in the G1, S, G2, and/or mitotic phase untreated and at 2, 6, 8, 18, and 24 h after irradiation. At 6 and 8 h post-irradiation, all the KO mES cells demonstrated an enlarged G2 phase population, with the largest increase observed in cells lacking Rad54 or both Rad54 and DNA-PK_cs_ (Figure 3). At the 8, 18, and 24 h marks following 1 or 2 Gy X-ray radiation, the DNA-PK_cs_^−/−^ Rad54^−/−^ mES cells still demonstrated a higher proportion of cells in the G2 phase compared to the WT, Rad54^−/−^, and DNA-PK_cs_^−/−^ mES cells (Figure 3). The WT and DNA-PK_cs_^−/−^ mES cell lines show a population of G1 and S phase cells at 6 h after irradiation with 1 or 2 Gy of X-ray, indicating that some cells recover from the G2 cell cycle block and resume cell cycle progression (Figure 3a,b). This recovered population is not observed at 6 h after treatment with 1 or 2 Gy of X-ray radiation in cells lacking Rad54 or both Rad54 and DNA-PK_cs_ (Figure 3a,b). However, at the 8 h point following 1 and 2 Gy of X-ray irradiation, a minor population of G1 and S phase cells also emerged in the Rad54^−/−^ cells (Figure 3). The recovery from the G2 cell cycle block is least efficient in DNA-PK_cs_^−/−^ Rad54^−/−^ cells, which is evident from the reduced percentage of cells in the G1 phase at 8 h and 18 h after 1 and 2 Gy of X-ray irradiation (Figure 3). This observation suggests that there is a delayed or less efficient recovery from the G2 cell cycle block in Rad54^−/−^ and DNA-PK_cs_^−/−^ Rad54^−/−^ mES cells compared to WT and DNA-PK_cs_^−/−^ mES cells and that this recovery is slowest in mES cells lacking both DNA-PK_cs_ and Rad54.

We measured the number of cells in mitosis by staining for phosphorylated histone H3 on serine 10 (phospho-H3). This marker correlates with chromatin condensation during the late G2 phase and prophase to telophase during mitosis [30,31]. Interestingly, we observed that the levels of phospho-H3 are higher in DNA-PK_cs_^−/−^ Rad54^−/−^ mES cells compared to WT, DNA-PK_cs_^−/−^, and Rad54^−/−^ mES cells at 24 h after 1 Gy of X-ray irradiation (Figure 3a,b). This observation indicates that these cells undergo progression from G2 to the mitotic phase at later time points post-irradiation. Immunofluorescence staining revealed that cells with enlarged nuclei are not positive for phospho-H3 (Appendix A). Nevertheless, phospho-H3-positive pro-, meta-, and anaphase stages were detected at 24 h after 1 or 2 Gy of X-ray radiation in the DNA-PK_cs_^−/−^ Rad54^−/−^ mES cells (Appendix A). This indicates that some cells lacking DNA-PK_cs_ and Rad54 indeed go through mitosis at later time points after X-ray irradiation. In conclusion, these data show that X-ray irradiation induces a G2 cell cycle block in mES cells lacking either Rad54 or DNA-PK_cs_ or both. The recovery from this cell cycle block is slower in mES cells lacking Rad54 compared to WT cells or cells lacking only DNA-PK_cs_. Moreover, cells lacking both Rad54 and DNA-PK_cs_ show an even more persistent G2 cell cycle block. However, at 18 and 24 h after X-ray irradiation, cells lacking DNA-PK_cs_ and Rad54 also seem to progress through mitosis.

### 3.4. mES Cells Lacking DNA-PK_cs_ and Rad54 or Expressing ATPase-Defective Rad54 Show Similar Sensitivity to X-Ray Radiation

To investigate the role of Rad54 in cells deficient in NHEJ, we generated DNA-PK_cs_^−/−^ mES cells that expressed GFP-fused WT Rad54 or GFP-fused Rad54 bearing a Lysine (K)-to-Arginine (R) mutation at amino acid position 189, resulting in ATPase-defective Rad54, from the endogenous locus (Figure 4a). We confirmed that the newly generated DNA-PK_cs_^−/−^ mES Rad54^WT-GFP/−^ cell lines lacked protein expression of DNA-PK_cs_ (Figure 4b). Moreover, we used immunoblotting and sequencing to show that the cells expressed Rad54^WT-GFP^ or Rad54^KR-GFP^ (Figure 4b,c). We subjected WT, Rad54^−/−^, DNA-PK_cs_^−/−^, Rad54^−/−^ DNA-PK_cs_^−/−^, WT Rad54^KR-GFP/−^, DNA-PK_cs_^−/−^ Rad54^WT-GFP/−^, and DNA-PK_cs_^−/−^ Rad54^KR-GFP/−^ mES cells to X-ray radiation and assessed their clonogenic survival. The DNA-PK_cs_^−/−^ mES cells and WT Rad54^KR-GFP/−^ mES cells showed 3.3- and 1.2-fold increased sensitivity compared to the WT mES cells, respectively (Figure 4d, Appendix A). Interestingly, cells both lacking DNA-Pk_cs_ and expressing Rad54^KR-GFP^ show 4.9-fold sensitivity to X-ray radiation and thus are hypersensitive to X-ray radiation (Figure 4d, Appendix A). We noticed that mES cells completely lacking Rad54 were slightly more sensitive in the clonogenic survival assay compared to cells expressing ATPase-defective Rad54. This implies that cells with Rad54 bearing a K189R mutation in the ATPase domain retains some functionality in repairing DSBs induced by X-ray radiation.

Next, we determined whether the increased sensitivity of the DNA-PK_cs_^−/−^ Rad54^KR-GFP/−^ mES cell lines coincides with the delayed resolution of 53BP1 foci and a persistent G2 cell cycle block after irradiation. Therefore, we studied the kinetics of 53BP1 focus formation and resolution and cell cycle progression following X-ray irradiation in WT Rad54^WT-GFP/−^, WT Rad54^KR-GFP/−^ DNA-PK_cs_^−/−^ Rad54^WT-GFP/−^, and DNA-PK_cs_^−/−^ Rad54^KR-GFP/−^ mES cells. We showed that the decrease in 53BP1 foci over time was slower in the DNA-PK_cs_^−/−^ Rad54^KR/−^ mES cells compared to the mES cells only lacking DNA-PK_cs_ or only expressing Rad54^KR^ (Appendix A). Furthermore, as observed before for the DNA-PK_cs_^−/−^ and DNA-PKcs^−/−^ Rad54^−/−^ mES cells, the DNA-PK_cs_^−/−^ Rad54^WT-GFP/−^ and DNA-PK_cs_^−/−^ Rad54^KR-GFP/−^ mES cells also showed a substantial increase in nuclear area at 18 and 24 h post X-ray irradiation (Figure 5a,c, Appendix A). When we analyzed the cell cycle progression of the WT Rad54^WT-GFP/−^, WT Rad54^KR-GFP/−^, DNA-PK_cs_^−/−^ Rad54^WT-GFP/−^, and DNA-PK_cs_^−/−^ Rad54^KR-GFP/−^ mES cells after 1 or 2 Gy of X-ray irradiation, we observed that at 6 and 8 h post-irradiation, mES cells lacking DNA-PK_cs_ or expressing ATPase-defective Rad54 or both had an enlarged G2 phase population (Appendix A). The largest increase in G2 phase cells was observed for the mES cells lacking DNA-PK_cs_ and expressing ATPase-defective Rad54 (Appendix A). In conclusion, these results demonstrate that DNA-PK_cs_^−/−^ Rad54^KR-GFP/−^ mES cells and DNA-PK_cs_^−/−^ Rad54^−/−^ mES cells have a similar phenotypic response to X-ray irradiation.

### 3.5. mES Cells Lacking DNA-PK_cs_ Show Elevated Levels of Rad54 Foci after X-ray Irradiation

To study the role of WT or ATPase-defective Rad54 in the absence of DNA-PK_cs_, we quantitated the number of Rad54 foci in untreated cells and cells irradiated with 2 Gy of X-ray at 2, 6, 8, 18, and 24 h post-irradiation. As shown before, in the absence of externally induced DNA damage, there was an elevated number of Rad54 foci within the ATPase-defective Rad54 mutant WT cells [11]. This increase in spontaneous Rad54 foci does not represent increased endogenous DNA damage. Our current study shows that within a DNA-PK_cs_ KO context, cells expressing ATPase-defective Rad54 also show an elevated number of Rad54 foci without the induction of DNA damage (Figure 5a,b, Appendix A). There is no major increase in 53BP1 foci in unchallenged DNA-PK_cs_^−/−^ Rad54^KR-GFP/−^ mES cells, suggesting that the spontaneous increase in Rad54 foci cannot be attributed to an increased level of unrepaired DSBs (Appendix A). The number of Rad54 foci increased after X-ray irradiation and decreased over time in all genotypes of the mES cells (Figure 5a,b, Appendix A). The number of Rad54 foci remained at higher levels over time in the WT Rad54^KR-GFP/−^, DNA-PK_cs_^−/−^ Rad54^WT-GFP/−^, and DNA-PK_cs_^−/−^ Rad54^KR-GFP/−^ mES cell lines compared to the WT Rad54^WT-GFP/−^ cell line (Figure 5a,b, Appendix A). As previously demonstrated, these results indicate that the ATPase activity of Rad54 affects its dissociation from foci [11]. Additionally, the findings suggest that in the absence of DNA-PK_cs_, HR becomes more important for the repair of IR-induced DSBs.

To gain more insight into the kinetics of Rad54 focus formation and resolution in mES cells lacking DNA-PK_cs_ and expressing WT or catalytically inactive Rad54, we performed live-cell imaging of the WT Rad54^WT-GFP/−^, WT Rad54^KR-GFP/−^_,_ DNA-PK_cs_^−/−^ Rad54^WT-GFP/−^, and DNA-PK_cs_^−/−^ Rad54^KR-GFP/−^ mES cells for 24 h after 2 Gy of X-ray irradiation in the presence of the DNA dye Spy650. We observed that mES cells lacking DNA-PK_cs_ do not undergo division often but develop enlarged nuclei after irradiation (Figure 6c,d). Additionally, we observed that the Rad54 foci disappear during mitosis (Figure 6a,b). Interestingly, the Rad54 foci also disappear in cells with enlarged nuclei (Figure 6a,b), indicating that these cells are likely at the boundary between the G2 phase and mitosis or in a very early phase of mitosis. However, the disappearance of Rad54 foci does not imply that DSB repair is completed and that the IR-induced DSBs are resolved in cells with enlarged nuclei. mES cells lacking DNA-PK_cs_ exhibit enlarged nuclei coinciding with the loss of Rad54 foci, resulting in a lower average number of Rad54 foci per nucleus in fixed samples in the DNA-PK_cs_^−/−^ Rad54^KR-GFP/−^ mES cells compared to the WT Rad54^KR-GFP/−^ mES cells. Therefore, we excluded enlarged nuclei (>400 µm^2^) from the quantification of Rad54 foci in the fixed samples, resulting in comparable levels of Rad54 foci in the WT and DNA-PK_cs_^−/−^ mES cells expressing ATPase-defective Rad54 (Figure 5b). Together, these results show that Rad54 foci levels following irradiation are affected by both the absence of DNA-PK_cs_ and the expression of ATPase-defective Rad54.

### 3.6. Increased Genomic Instability in Cells Lacking DNA-PK_cs_ and Expressing ATPase-Defective Rad54

During live-cell imaging, we not only observed cells with enlarged nuclei but we also noted that some of the DNA-PK_cs_^−/−^ Rad54^WT-GFP/−^ and DNA-PK_cs_^−/−^ Rad54^KR-GFP/−^ mES cells were able to divide at later time points following irradiation. However, cell division of these cells often results in the formation of micronuclei. Cells with micronuclei are also visible in the samples fixed at 24 h after irradiation (Figure 7a). To gain insight into the extent of micronuclei formation in the WT Rad54^WT-GFP/−^, WT Rad54^KR-GFP/−^_,_ DNA-PK_cs_^−/−^ Rad54^WT-GFP/−^, and DNA-PK_cs_^−/−^ Rad54^KR-GFP/−^ mES cells, we quantified the percentage of cells with micronuclei and the number of micronuclei per cell in fixed cells 24 h post-irradiation, as well as in live cells imaged for 24 h in the presence of the DNA dye Spy650. We observed a slight increase in the percentage of cells with micronuclei after 2 Gy of X-ray radiation in the WT Rad54^KR-GFP/−^ mES cells compared to WT Rad54^WT-GFP/−^ (Figure 7a,b). The percentage of cells with micronuclei after 2 Gy of X-ray radiation was even higher in the DNA-PK_cs_^−/−^ Rad54^WT-GFP/−^ mES cells (Figure 7a,b). Moreover, 2 Gy of X-ray irradiation of mES cells both lacking DNA-PKcs and expressing Rad54^KR^ results in a very high percentage (~60–80%) of cells with micronuclei (Figure 7a,b). Interestingly, the number of micronuclei per cell is also higher in cells lacking DNA-PK_cs_ and expressing ATPase-defective Rad54 (Figure 7c). In addition to the micronuclei, we quantified the percentage of cells with chromatin bridges in the live-cell imaging data. We observed chromatin bridges in all genotypes of the mES cells (Figure 7b). The percentage of cells with chromatin bridges appears to be slightly higher in the DNA-PK_cs_^−/−^ Rad54^KR-GFP/−^ mES cells (Figure 7b). These results show that mES cells deficient in both NHEJ and HR exhibit increased genomic instability after X-ray irradiation.

## 4. Discussion

This study underscores the complex interplay between DNA-PK_cs_ and Rad54 in the repair of IR-induced DSBs in mES cells. Our results demonstrate that simultaneous deficiency of DNA-PKcs and Rad54, or DNA-PK_cs_ deficiency and the expression of ATPase-defective Rad54, renders mES cells hypersensitive to X-ray radiation. This sensitivity highlights the indispensable roles of both the NHEJ and HR pathways in safeguarding genomic integrity from IR-induced DNA damage.

### 4.1. Increased Nuclear Size of mES Cells Lacking DNA-PK_cs_

We observed that mES cells lacking DNA-PK_cs_ exhibit an increased nuclear size at later time points (~18–24 h) post-irradiation. Quantification of the DNA content revealed that the cells with larger nuclei had an elevated DNA content compared to cells with smaller nuclei, indicating that the cells with enlarged nuclei are G2 phase cells. Live-cell imaging showed that Rad54 foci disappear during mitosis, similar to other DNA repair proteins like BRCA1 and 53BP1 [32]. The fact that we also see the disappearance of Rad54 foci in cells with enlarged nuclei indicates that these cells might be in a very early stage of mitosis or be approaching mitosis. This arrest at the G2/M boundary or in early mitosis could be due to DNA-PK_cs_ deficiency, which delays the prometaphase-to-anaphase transition [33,34]. Another explanation for the nuclear enlargement in irradiated DNA-PK_cs_^−/−^ mES cells could be RPA exhaustion [35]. Given that HR acts on resected DNA ends and the Rad54 foci levels remain elevated in DNA-PK_cs_^−/−^ mES cells, increased resection likely occurs in these cells. mES cells also experience increased replication stress, characterized by ssDNA gap accumulation and extensive fork reversal, both of which involve RPA accumulation [36]. Thus, high levels of resected DSBs in NHEJ- and HR-deficient mES cells, combined with replication stress, may lead to RPA exhaustion, resulting in nuclear enlargement. Furthermore, chromatin decondensation as a consequence of a high DNA damage load may contribute to nuclear enlargement in irradiated mES cells lacking DNA-PK_cs_ [37].

### 4.2. The Role of DNA-PK_cs_ and Rad54 in Cell Cycle Regulation and Genome Stability Maintenance in mES Cells

DNA damage induced by IR triggers strong G2 cell cycle arrest in mES cells lacking DNA-PK_cs_ and Rad54, with no observed G1 cell cycle arrest, consistent with the lack of a G1/S checkpoint in mES cells [38,39,40]. These cells rely on the G2/M checkpoint and an extended S phase, which promotes error-free repair through HR, to maintain genome stability. The G2 arrest is more persistent in Rad54^−/−^ mES cells than in DNA-PK_cs_^−/−^ cells, likely due to the less efficient repair of DSBs induced during the S phase in Rad54^−/−^ cells. The most persistent G2 arrest was observed in mES cells lacking both DNA-PK_cs_ and Rad54, although this was temporary, with the cells progressing through mitosis at later time points. The G2/M checkpoint has intrinsic insensitivity, only responding to a level of 10–20 DSBs and terminating arrest before repair is complete [41,42]. Furthermore, a study in yeast showed that the G2/M checkpoint is blind to replication and recombination intermediates [43]. Thus, it is possible that cells enter mitosis with unresolved DSBs and HR intermediates, such as heteroduplexes and unresolved Holliday junctions, which might lead to genomic instability upon cell division. This is particularly true in NHEJ- and HR-deficient mES cells, where the absence of Rad54 or the presence of catalytically inactive Rad54 could result in the accumulation of HR intermediates [44]. Additionally, our findings suggest that DNA-PK_cs_-deficient cells rely more on HR, contributing to more recombination intermediates during division.

Cell division with unresolved DSBs can result in micronuclei formation [45]. We observed increased levels of micronuclei in the mES cells following irradiation, especially in those lacking DNA-PK_cs_, consistent with the role of DNA-PK_cs_ in mitotic progression and chromosomal alignment [33]. Additionally, other studies have shown increased micronuclei formation in NHEJ-deficient cells after irradiation [46,47]. Our findings reveal that micronuclei formation is further elevated in cells lacking both Rad54 and DNA-PK_cs_. This suggests increased DSB levels at the time of division in these cells, which is likely due to impaired DSB repair mechanisms. These findings demonstrate that deficiencies in both NHEJ and HR lead to increased genomic instability upon IR treatment.

Altogether, we demonstrate that cells deficient in HR and NHEJ are hypersensitive to X-ray radiation, coinciding with impaired DSB repair, persistent G2 cell cycle block, and increased genomic instability. In WT mES cell, the mutagenic Theta-Mediated End Joining (TMEJ) repair pathway plays a minor role in repairing IR-induced breaks [48]. Our results indicate that in the surviving mES cells lacking DNA-PK_cs_ and Rad54 or expressing ATPase-defective Rad54, DSBs are repaired by TMEJ. Recently, it was shown that TMEJ primarily functions during the mitotic phase of the cell cycle [49,50,51]. Our findings indicate that mES cells deficient in HR and NHEJ are arrested in the G2 phase, which may explain why TMEJ does not compensate for DSB repair in these cells and why they exhibit hypersensitivity to IR.

### 4.3. Enhanced Activity of Rad54 in the Absence of DNA-PK_cs_

To understand the role of Rad54 in the absence of DNA-PK_cs_, we generated DNA-PK_cs_^−/−^ mES cells endogenously expressing GFP-tagged Rad54^WT^ or Rad54^K189R^. Cells expressing Rad54^K189R^ are slightly less sensitive to X-ray radiation compared to cells completely lacking Rad54, indicating some retained functionality in DSB repair. Previous research from our lab has already shown that WT Rad54^K189R-GFP/−^ mES cells are slightly less sensitive to photons compared to Rad54^−/−^ mES cells [11]. In contrast, WT Rad54^K189A-GFP/−^ showed the same sensitivity to photons as Rad54^−/−^ cells [11]. We attempted to create DNA-PK_cs_^−/−^ Rad54^K189A-GFP/−^ mES cells but failed, likely due to the lower gene targeting efficiency in these cells.

As previously demonstrated in WT mES cells, DNA-PK_cs_^−/−^ mES cells expressing ATPase-defective Rad54 show an increased number of Rad54 foci in unirradiated cells [11]. There is no corresponding increase in the number of 53BP1 foci in these cells, suggesting that the increase in Rad54 foci is not caused by an increase in endogenous DSBs. However, because 53BP1 plays a role in protecting DSB ends from end resection, 53BP1 foci may be more prominent during repair by NHEJ, which acts on non-resected ends, and less visible when DSBs are repaired by HR [52]. To confirm that the increased number of Rad54 foci is not due to elevated DSB levels, alternative DSB detection methods, such as the neutral comet assay, could be employed. Quantification of the Rad54 foci showed elevated levels post-irradiation in the DNA-PK_cs_^−/−^ Rad54^WT-GFP/−^ cells compared to the levels in WT Rad54^WT-GFP/−^ mES cells, suggesting increased HR activity. Previous research indicates that the DNA-PK complex, consisting of Ku proteins and DNA-PK_cs_, acts as a ‘gatekeeper’ regulating DNA end access [53,54]. This suggests that in the absence of DNA-PK_cs_, end resection might be increased, resulting in DSB intermediates that can be repaired by HR. In addition to assessing the Rad54 foci levels at fixed time points after irradiation, we employed live-cell imaging. We observed that mES cells lacking DNA-PK_cs_ are arrested in their cell cycle and exhibit enlarged nuclei, which coincides with the loss of Rad54 foci. Together, our results show that Rad54 foci levels are influenced by a lack of DNA-PK_cs_ and the expression of ATPase-defective Rad54, as well as cell cycle progression and arrest.

### 4.4. Therapeutic Applications

Our data suggest that inhibiting NHEJ, HR, or both in combination with radiation may have therapeutic benefits. We used mES cells because they are naturally immortalized and do not undergo mutagenic changes during in vitro culture. This genetic stability makes them ideal for studying the effects of DNA-PK_cs_ and Rad54 deletion without the confounding influence of additional mutations. However, it would also be important to test the response to X-ray radiation in NHEJ- and HR-deficient cancer cells. Additionally, using chemical inhibitors of NHEJ or HR would offer valuable insights into the potential to implement NHEJ or HR inhibition in clinical settings. Currently, various DNA-PK_cs_ inhibitors are available and have been investigated in preclinical studies. These studies have demonstrated enhanced sensitivity when radiation is combined with DNA-PK_cs_ inhibitors [55,56,57,58,59]. Moreover, a few clinical trials evaluating the effects of DNA-PK_cs_ inhibition and radiation on efficacy and safety in humans are underway (e.g., NCT03770689, NCT05116254). Recently, results from a phase 1 clinical trial (NCT02516813) using the DNA-PK_cs_ inhibitor peposertib were published. The results show that peposertib is a potent radiosensitizer, although with a narrow therapeutic window [60]. Our findings indicate that the use of DNA-PK_cs_ inhibitors offers a promising approach to improving the efficacy of radiotherapy in cancer treatment. Additionally, our findings suggest that DNA-PK_cs_ inhibitors in combination with radiotherapy could be particularly beneficial in HR-deficient tumors.

## 5. Conclusions

Together, these findings highlight the intricate interplay between NHEJ and HR in maintaining genomic stability upon IR-induced DSBs. Understanding the specific roles and interactions of DNA-PK_cs_ and Rad54 provides critical insights into the repair mechanisms and their implications for cell cycle dynamics, genomic integrity, and potential therapeutic strategies.

## Figures and Tables

**Figure 1 cells-13-01462-f001:**
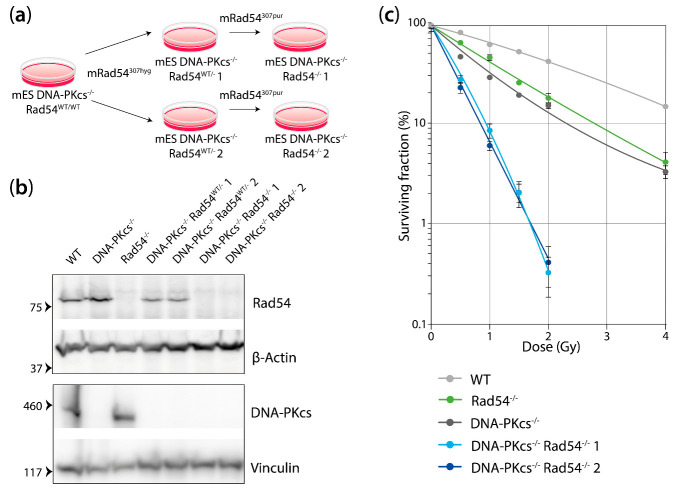
mES cells lacking DNA-PK_cs_ and Rad54 are hypersensitive to X-ray radiation. (**a**) Two independent clones of the DNA-PK_cs_^−/−^ Rad54^−/−^ mES cell line were generated in a two-step process. First, DNA-PK_cs_^−/−^ mES cells were targeted with a targeting construct against Rad54 containing a hygromycin resistance gene to generate DNA-PK_cs_^−/−^ Rad54^WT/−^ mES cells. Secondly, two independent DNA-PK_cs_^−/−^ Rad54^WT/−^ mES cell clones were targeted with a targeting construct against Rad54 containing a puromycin resistance gene to generate DNA-PK_cs_^−/−^ Rad54^−/−^ mES cells. (**b**) Western blot was used to confirm the lack of Rad54 and DNA-PK_cs_ in the mES cells with the indicated genotypes. The upper Western blot in the figure shows probing for Rad54 and β-actin as the loading control. The lower Western blot in the figure shows probing for DNA-PK_cs_ and vinculin as the loading control. (**c**) Clonogenic survival of mES cell lines with indicated genotypes after X-ray irradiation. Error bars represent SEM.

**Figure 2 cells-13-01462-f002:**
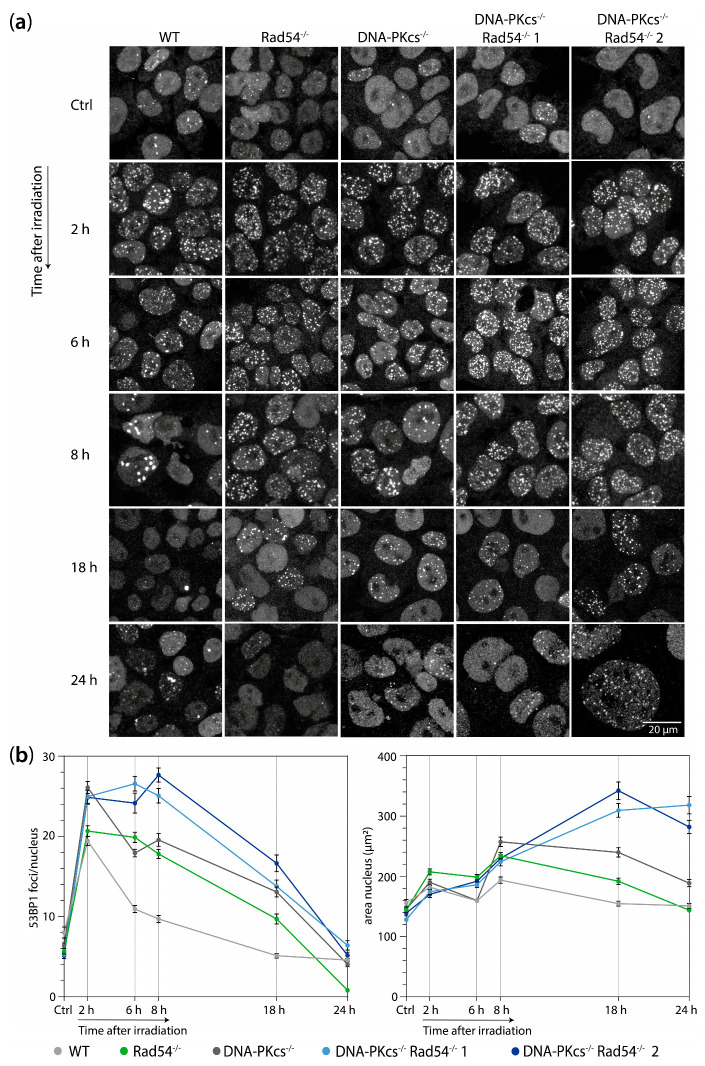
mES cells lacking DNA-PK_cs_ and Rad54 show impaired resolution of 53BP1 foci and increased nuclear size after 2 Gy of X-ray radiation. (**a**) Representative images of mES cells irradiated with 2 Gy of X-ray radiation and incubated for indicated times. After the recovery time, cells were fixed and stained for 53BP1. (**b**) Quantification of 53BP1 foci per mES nucleus (left) and area of mES nuclei (right). Error bars represent SEM.

**Figure 3 cells-13-01462-f003:**
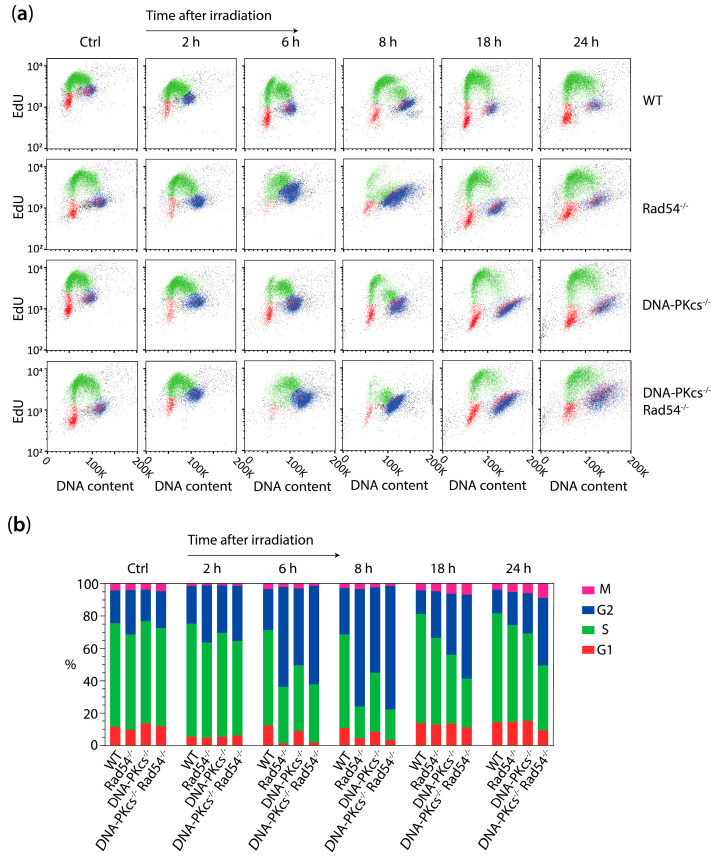
X-ray irradiation results in more persistent G2 phase cell cycle block in mES cells lacking DNA-PK_cs_ and Rad54. (**a**) mES cells were irradiated with 1 Gy of X-ray radiation and incubated for indicated times. After recovery time, cells were fixed and stained for DAPI (DNA content), EdU (S phase cells), and phospho-H3 (mitotic (M) phase cells). Cell cycle distribution was analyzed using flow cytometry. (**b**) Quantification of percentage of G1, S, G2, and M phase cells in mES cells irradiated with 1 Gy of X-ray radiation, as shown in (**a**). (**c**) mES cells were irradiated with 1 and 2 Gy of X-ray radiation and incubated for indicated times. After recovery time, cells were fixed, and DNA was stained using Propidium Iodide. Cell cycle distribution was analyzed using flow cytometry. (**d**) Quantification of percentage of G1, S, and G2 phase cells in mES cells irradiated with 1 and 2 Gy of X-ray radiation, as shown in (**c**).

**Figure 4 cells-13-01462-f004:**
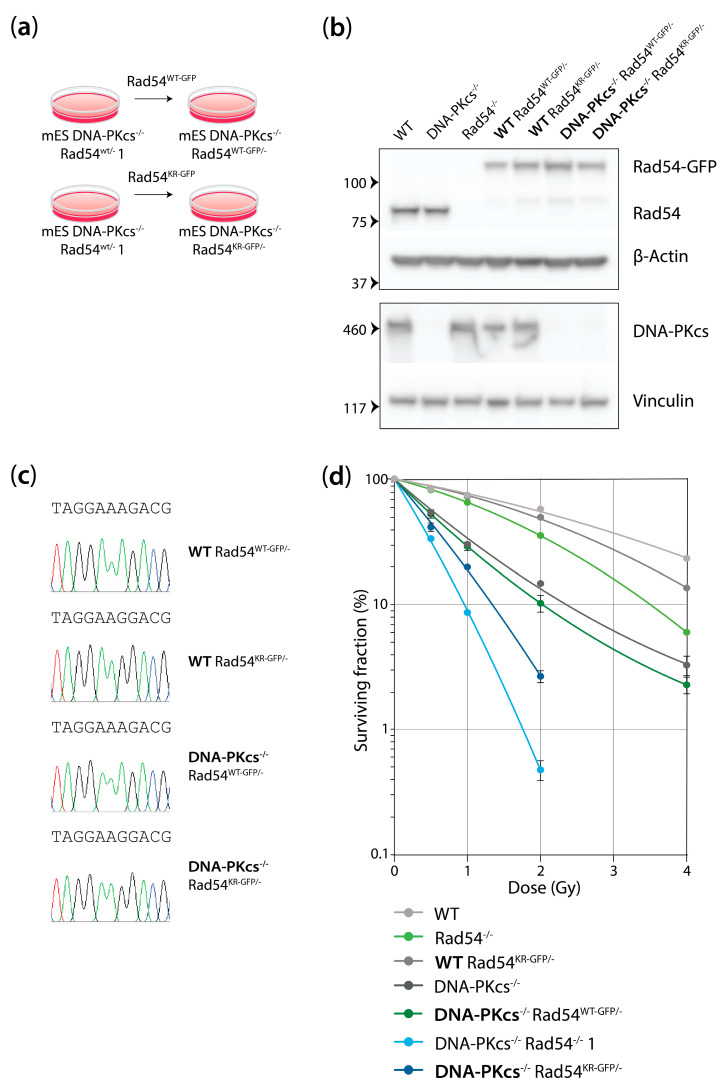
mES cells lacking DNA-PK_cs_ and expressing ATPase-defective Rad54 are hypersensitive to X-ray radiation. (**a**) The DNA-PK_cs_^−/−^ Rad54^WT-GFP/−^ and DNA-PK_cs_^−/−^ Rad54^KR-GFP/−^ mES cell lines were generated by targeting the mES DNA-PK_cs_^−/−^ Rad54^WT/−^ 1 cell line with a targeting construct against Rad54 containing Rad54^WT-GFP^ or Rad54^KR-GFP^. (**b**) Western blot was used to confirm the knockin of GFP-Rad54 in mES cells with the indicated genotypes. The upper Western blot in the figure shows probing for Rad54 and β-actin as the loading control. The lower Western blot in the figure shows probing for DNA-PKcs and vinculin as the loading control. (**c**) Sanger sequencing results to confirm K189R mutation in Rad54. (**d**) Clonogenic survival of mES cell lines with indicated genotypes after X-ray irradiation. Error bars represent SEM.

**Figure 5 cells-13-01462-f005:**
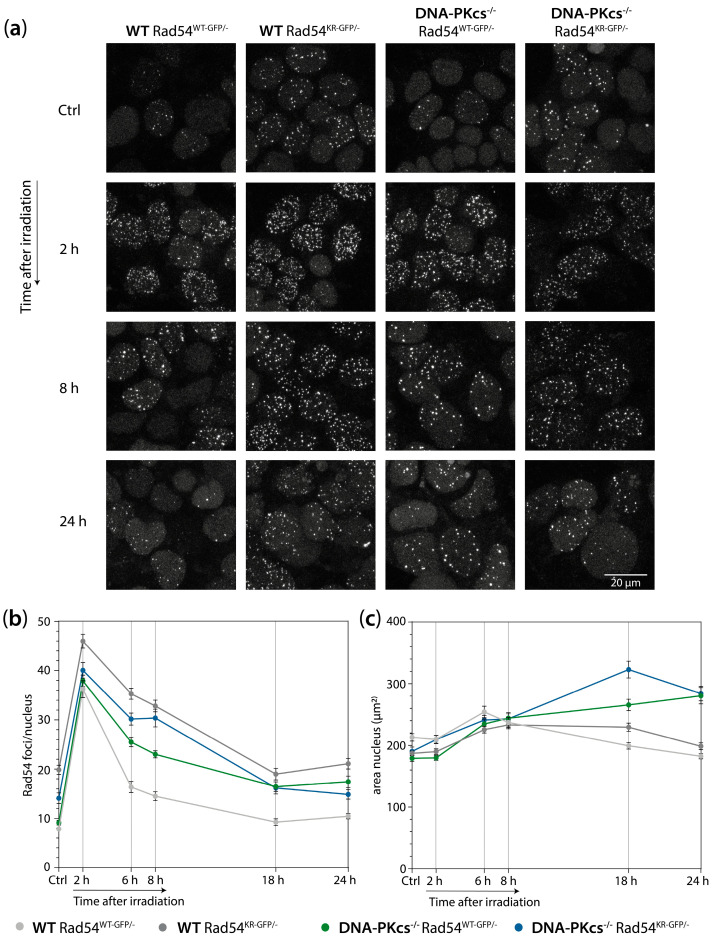
mES cells lacking DNA-PK_cs_ show impaired Rad54 focus resolution and increased nuclear size after 2 Gy of X-ray radiation. (**a**) Representative images of mES cells irradiated with 2 Gy of X-ray radiation and incubated for indicated times. After the recovery time, cells were fixed and imaged for Rad54. (**b**) Quantification of Rad54 foci per mES nucleus. Rad54 foci disappear in cells with enlarged nuclei (Figure 6); therefore, nuclei larger than 400 µm^2^ were excluded from the analysis. Error bars represent SEM. (**c**) Quantification of area of mES nuclei. Error bars represent SEM.

**Figure 6 cells-13-01462-f006:**
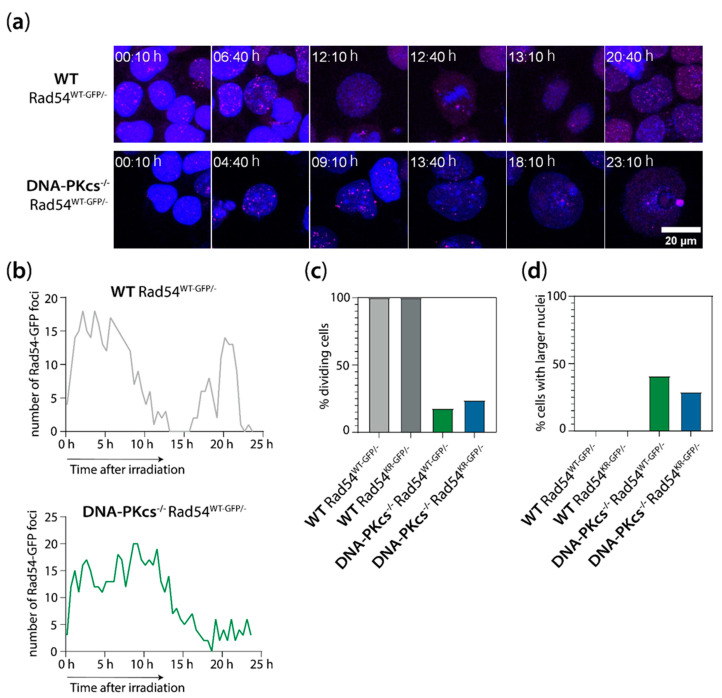
Rad54 foci disappear in mitotic cells and in cells with enlarged nuclei. (**a**) Representative images of live-cell imaging of WT Rad54^WT-GFP/−^ (top) and DNA-PK_cs_^−/−^ Rad54^WT-GFP/−^ (bottom) mES cells. Top row shows the cells going through cell division with the disappearance of Rad54 foci towards cell division. Bottom row shows the cell with disappearing Rad54 foci, swelling up but with no cell division happening. (**b**) Quantification of the Rad54-GFP foci in the cells shown in (**a**). (**c**) Quantification of the percentage of dividing cells in the live-cell image. (**d**) Quantification of the percentage of cells that show a larger nuclear size.

**Figure 7 cells-13-01462-f007:**
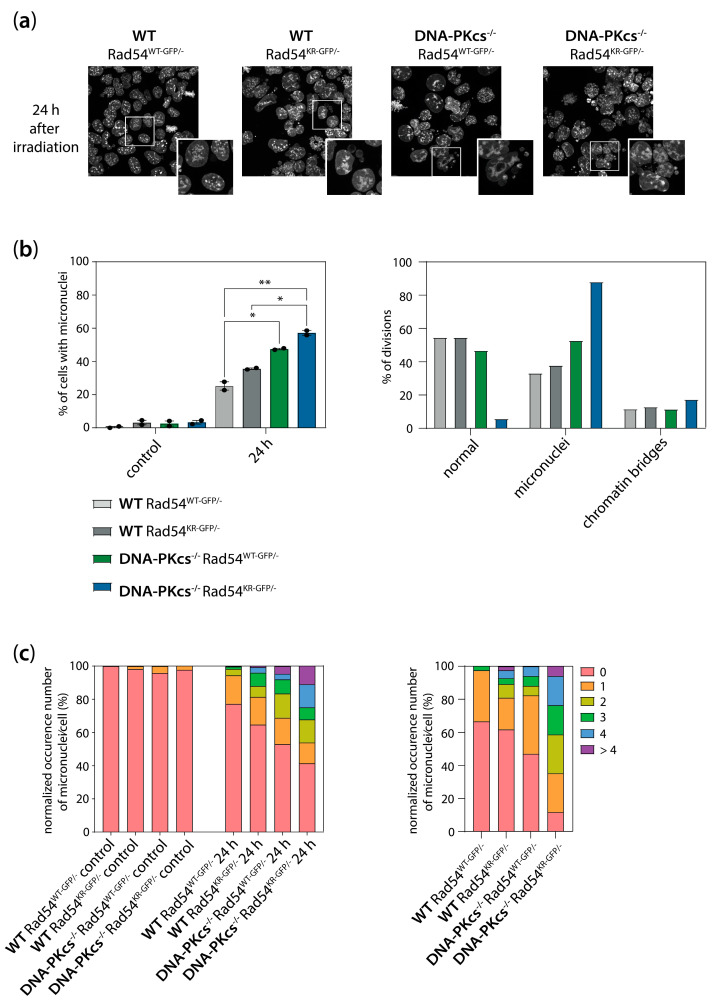
Increased micronuclei formation in mES cells lacking DNA-PK_cs_ and expressing ATPase-defective Rad54. (**a**) Representative DAPI images of WT Rad54^WT-GFP/−^, WT Rad54^KR-GFP/−^, DNA-PK_cs_^−/−^ Rad54^WT-GFP/−^, and DNA-PK_cs_^−/−^ Rad54^KR-GFP/−^ mES cells 24 h after 2 Gy of X-ray radiation. (**b**) Quantification of the percentage of cells that have micronuclei in fixed samples. Error bars represent SEM. Asterisks represent the following *p*-values: * ≤ 0.05; ** ≤ 0.01 (left). Quantification of the percentage of cells that are normal, have micronuclei, or have chromatin bridges in live-cell imaging data (right). (**c**) Number of micronuclei per cell in fixed samples (left, normalized for total number of nuclei) and live-cell imaging data (right, normalized for total number of cell divisions).

## Data Availability

The raw data supporting the conclusions of this article will be made available by the authors on request.

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
