# Peer review of "Synergistic Roles of Non-Homologous End Joining and Homologous Recombination in Repair of Ionizing Radiation-Induced DNA Double Strand Breaks in Mouse Embryonic Stem Cells"

_cells, 2024, doi:10.3390/cells13171462_

Round 1

Reviewer 1 Report

Comments and Suggestions for Authors

The authors used mouse embryonic stem (mES) cells to generate knockout (KO) mutants for DNA-PKcs and Rad54 to explore the combined role of HR and NHEJ in DSB repair. They report that cells lacking both proteins, DNA-PKcs and RAD54, are hypersensitive to X-ray radiation, and show impaired 53BP1 focus resolution suggestive of impaired repair of DNA DSBs and a more persistent G2 phase cell cycle block, which is also in line with reduced DNA DSB repair efficiency. The authors further report that mES cells deficient in DNA-PKcs, or both DNA-PKcs and Rad54 exhibit increased nuclear size 18-24 hours post-irradiation. Notably, to obtain further mechanistic insights into the role of Rad54 in the absence of DNA-PKcs, the authors generated DNA-PKcs KO mES mutants expressing GFP-tagged wild-type (WT) or ATPase-defective RAD54. They tracked Rad54 foci over time post-irradiation and report that cells lacking DNA-PKcs and expressing ATPase-defective Rad54 exhibit a similar phenotypic response to IR as mutants lacking both DNA-PKcs and Rad54. In addition, and despite a strong G2 phase arrest, live-cell imaging uncovers that double mutants eventually progress through mitosis, but show increased incidence of micronuclei formation. It is also interesting that the authors find that mES cells lacking DNA-PKcs display increased Rad54 foci, which suggests enhanced engagement on HR.

The reported results together confirm essential roles of HR and NHEJ in maintaining genomic stability in irradiated mES cells and suggest a well-tuned up interplay between the main pathways of DSB repair that has direct consequences to their progression through the cell cycle. The paper is well written and conveys its message in a clear way.

Some minor point are as follows:

1. In Materials and Methods, lines 95-10, the authors should explain whether numbers refer to amounts for 1 L of DMEM.

2. Add the units of time (h) in Figure 6.

Reviewer 2 Report

Comments and Suggestions for Authors

This study examines the interaction between DNA-PKcs and Rad54 in repairing double-strand breaks (DSBs) caused by ionizing radiation (IR) in mouse embryonic stem (mES) cells. The authors generate DNA-PKcs-/- Rad54-/-  mES cells that express either wild-type Rad54 or Rad54 with an ATPase-defective mutation (Rad54K189R) and assess their sensitivity to DNA damage and repair efficiency. The results showed that cells lacking DNA-PKcs and expressing ATPase-defective Rad54 exhibited the highest sensitivity to X-ray radiation. Rad54K189R retained partial functionality, as cells with this mutant showed some repair capability compared to cells completely lacking Rad54. Additionally, DNA-PKcs-/- and Rad54K189R-expressing cells had delayed resolution of 53BP1 foci, increased nuclear size, and an accumulation of cells in the G2 phase, highlighting defects in DNA damage response and cell cycle regulation. Furthermore, increased genomic instability was observed, with elevated micronuclei and chromatin bridges, particularly in cells deficient in both DNA-PKcs and functional Rad54. Overall, these findings emphasize the critical roles of both NHEJ and HR in maintaining genomic stability and suggest potential implications for targeting DNA repair pathways in therapeutic strategies. To provide a more thorough investigation of the roles and mechanisms of DNA-PKcs and Rad54 in DNA repair, the following analyses could be incorporated:

1. The study focuses on mES cells, which may not fully represent the behavior of other cell types, especially cancer cells.

2. The research primarily examines the response to X-ray radiation. Including other types of DNA damage or stresses could provide a more comprehensive view of the roles of DNA-PKcs and Rad54 in DNA damage response.

3. Besides knockdown and rescue experiments, the authors could perform rescue experiments with small molecules or inhibitors. The authors could use chemical inhibitors or activators of related repair pathways to assess their impact on the repair processes in cells deficient in DNA-PKcs or Rad54.

4. Besides clonogenic survival assays, have authors examined cell proliferation, apoptosis, and dose response to damaging reagents?

5. Include more detailed mechanisms about crosstalk between NHEJ and HR pathways, especially molecular interactions between DNA-PKcs and Rad54 and downstream targets.

6. Determine if introducing functional forms of DNA-PKcs can rescue the observed defects. The authors could also express wild-type and mutant forms of DNA-PKcs in Rad54-/- cells and assess repair efficiency, foci formation, and cellular sensitivity to DNA damage.

7. Could other DNA repair pathways (alternative NHEJ, BER, NER) be involved?

8. missing p-values and no statistical analyses have been shown in the figures.

9. Line 97, should be 10% FCS, 1% non-essential amino acids.

Reviewer 3 Report

Comments and Suggestions for Authors

In the manuscript submitted by Gerarda van de Kamp et al., the authors report the synergistic roles of NHEJ and HR in post-ionizing radiation DSB repair in mouse ES cells. Although the synergistic roles of NHEJ and HR in DSB repair have been extensively reported and are well accepted in the field, this paper intriguingly explores the kinetics of DNA damage repair, cell cycle arrest, and mitosis at different time points after IR irradiation. They demonstrate that DSB repair is much slower in mES cells with both NHEJ and HR double deficiency compared to either WT or single deficiency, leading to more DSBs and enlarged nuclei at later time points. Mitosis is also most delayed in NHEJ and HR double-deficient mES cells. Even though these cells can go through mitosis, it results in significant genomic instability, marked by an increased number of micronuclei. However, the differential impact of NHEJ and HR at different cell cycle phases and at early versus late repair phases was not well discussed in the manuscript. I have several suggestions that could further increase the impact of this paper.

Major Comments:

1. In Figure 2, there are more 53BP1 foci in DNA-PKcs-/- and DNA-PKcs-/-Rad54-/- cells compared to either WT or Rad54-/- at 2 hours after irradiation. Could this be attributed to the sequential recruitment of NHEJ and HR factors to DSBs? Does NHEJ act more quickly at the early repair stage? Please comment and discuss this in the manuscript.

2. The authors demonstrated that the spontaneous increase of Rad54 foci (Fig. 5) cannot be attributed to an increase in DSBs as they found no major increase in 53BP1 foci. However, the absence of 53BP1 foci does not necessarily indicate a low level of double-strand breaks (DSBs). 53BP1 is more prominent in NHEJ. If a cell is favoring HR, 53BP1 foci might be less visible. 53BP1 foci formation can vary with the cell cycle phase; for example, it is more prominent in the G1 phase and less so in the S and G2 phases when HR is more active. To measure the DSBs, the authors could perform a neutral comet assay at pre-irradiation and 24 hours post-irradiation.

3. The authors found that Rad54 foci disappeared in cells with enlarged nuclei, which may lower the average number of Rad54 foci in cell lines with more enlarged nuclei. It would be worth excluding the enlarged nuclei while quantifying the Rad54 foci in Figure 5.

Minor Comments:

1. Line 48: "Blunt ends can be repaired by NHEJ."

2. Line 73-74: “SCID Rad54-/- double mutant mice” is confusing. What does this mean?

Round 2

Reviewer 3 Report

Comments and Suggestions for Authors

The authors have fully addressed my concerns and suggestions. I recommend this manuscript for publication.